

# Exploring tumor endothelial cells heterogeneity in hepatocellular carcinoma: insights from single-cell sequencing and pseudotime analysis

Jiachun Sun[1], Shujun Zhang[2], Yafeng Liu[2], Kaijie Liu[2] and Xinyu Gu[1]

[1] Department of Oncology, The First Affiliated Hospital, College of Clinical Medicine, Henan University of Science and Technology, Luoyang, China
[2] Department of Infectious Diseases, The First Affiliated Hospital, College of Clinical Medicine, Henan University of Science and Technology, Luoyang, China

Corresponding author
Xinyu Gu, hkdguxy@163.com

## ABSTRACT

**Objective:** This study aimed to explore the heterogeneity of tumor endothelial cells (TECs) in hepatocellular carcinoma (HCC) and their role in tumor progression, with the goal of identifying new therapeutic targets and strategies to improve patient prognosis.

**Methods:** Single-cell RNA sequencing data from nine primary liver cancer samples were analyzed, obtained from the Gene Expression Omnibus (GEO) database. Data preprocessing, normalization, dimensionality reduction, and batch effect correction were performed based on the Seurat package. HCC cell types were identified using uniform manifold approximation and projection (UMAP) and cluster analysis, and the different cell types were annotated using the CellMarker database. Pseudotime trajectory analysis was conducted with Monocle to explore the differentiation trajectory of TECs. MAPK signaling pathway activity and copy number variations (CNV) in TECs were analyzed in conjunction with data from The Cancer Genome Atlas (TCGA), the trans-well and wound healing assay was used for cell invasion and migration activity assessment.

**Results:** Two subgroups of TECs (TECs 1 and TECs 2) were identified, exhibiting distinct functional activities and signaling pathways. Specifically, TECs 1 may be involved in tumor cell proliferation and inflammatory responses, whereas TECs 2 is not only involved in cell proliferation pathways, but also enriched in pathways such as metabolic synthesis. Pseudotime analysis revealed dynamic changes in TECs subgroups during HCC progression, correlating specific gene expressions (such as PDGFRB, PGF, JUN, and NR4A1). Subsequently, the JUN gene was predicted by performing binding sites and was shown to act as a transcription factor that may regulate the expression of the PGF gene. CNV analysis highlighted key genes and pathways in TECs that might influence HCC progression, and the PGF as key regulatory factor mediated cell proliferation and migration.

**Conclusion:** The study revealed the heterogeneity of TECs in HCC and their potential roles in tumor progression, offering new perspectives and potential therapeutic targets for HCC molecular mechanisms. The findings emphasize the importance of further exploring TECs heterogeneity for understanding HCC pathogenesis and developing personalized treatment strategies.

# INTRODUCTION

Hepatocellular carcinoma (HCC), the predominant primary liver cancer, stands as the third leading cause of cancer-related death worldwide (*Sung et al., 2021*). With its high malignancy, HCC presents a less than 18% 5-year survival rate (*Villanueva, 2019*). The therapeutic arsenal for HCC spans liver resection, transplantation, image-guided ablation, arterial interventions, chemotherapy, and molecular-targeted therapies. Typically, a multimodal approach is adopted in the clinical management of HCC patients. Nonetheless, the propensity for high recurrence post-tumor eradication continues to compromise the treatment efficacy in advanced HCC stages (*Llovet et al., 2021*; *Bouattour, Raymond & Faivre, 2017*). Immunotherapy has emerged as a promising advancement, enhancing clinical outcomes in advanced HCC by activating the patient's immune response against the tumor, diverging from conventional treatment modalities (*Ringelhan et al., 2018*; *Gao et al., 2023*). Despite these advancements, the treatment of advanced HCC remains fraught with challenges due to the disease's resistance to current therapies and the adverse influence of concurrent liver pathologies on treatment outcomes. The ongoing evolution in HCC treatment highlights the imperative for continued research aimed at devising more efficacious therapeutic strategies, particularly for patients with limited options in advanced disease stages (*Yang et al., 2023*; *Adhoute et al., 2017*). While systemic therapy advancements for HCC, especially with the advent of targeted and immunotherapeutic approaches, have been notable, significant hurdles persist. These challenges encompass overcoming therapeutic resistance, managing the underlying liver disease in HCC patients, and broadening the spectrum of effective treatments for patients with advanced-stage disease. Current research and clinical trials are geared towards addressing these challenges, intending to ameliorate overall patient prognoses and foster more individualized therapeutic approaches.

Single-cell sequencing technology has substantially advanced the domain of cancer research, delivering profound insights into the cellular and molecular heterogeneity of tumors, inclusive of HCC. This approach enables the exploration of genomic, transcriptomic, and epigenomic landscapes at single-cell resolution, offering a more nuanced and comprehensive understanding of tumor biology over conventional bulk sequencing methods (*Lei et al., 2021*; *Hao et al., 2022*; *Zulibiya et al., 2023*). In the realm of cancer research, single-cell sequencing elucidates the intricacies of the tumor microenvironment, clarifies the processes of tumorigenesis, metastasis, and resistance to therapy, and identifies novel targets for treatment. Pertinently, in HCC, it facilitates a nuanced characterization of intratumoral heterogeneity, unveiling diverse cancer cell subpopulations each with distinct genetic and phenotypic signatures. This granularity is pivotal for developing targeted therapeutic strategies and refining prognostic models, considering the complexity of these tumors. The integration of single-cell sequencing data

with other omic layers, such as DNA methylation and chromatin accessibility, enhances our comprehension of the regulatory mechanisms underpinning cancer cell diversity and adaptability, shedding light on how these elements contribute to tumor evolution and heterogeneity (*Tian & Li, 2022*). Summarily, single-cell sequencing stands as a transformative tool in cancer research, especially within the HCC context, promising the identification of novel biomarkers and therapeutic targets, thus paving the pathway towards more personalized and effective cancer treatment methodologies (*Bian et al., 2018*).

Here, this research profoundly investigates the heterogeneity of tumor endothelial cells (TECs) in HCC based on scRNA-seq technology, uncovering the potential mechanisms through which TECs influence HCC progression. By identifying and characterizing cell populations, analyzing genomic variations, examining TEC subgroups, and conducting pseudotime analyses, a deeper understanding of cellular heterogeneity within the HCC microenvironment has been garnered. In conclusion, our study is expected to develop more effective therapeutic tools for HCC targeted therapy through a deeper understanding and targeted regulation of specific pathways and cell populations.

## MATERIALS AND METHODS

### Single-cell data acquisition and analysis

Single-cell datasets comprising nine primary liver cancer samples were downloaded from the Gene Expression Omnibus (GEO, https://www.ncbi.nlm.nih.gov/geo/) repository (GSE125449) at https://www.ncbi.nlm.nih.gov/geo/. The Read10X function from the Seurat package (*Stuart et al., 2019*) was utilized to import the raw data, retaining cells with 20% mitochondrial genes and a gene count ranging between 200 and 6,000. Data normalization was performed using the NormalizeData function. Post-PCA dimensionality reduction, the Harmony package (*Korsunsky et al., 2019*) was employed to mitigate batch effects between samples, with parameters set to max.iter.harmony = 20 and lambda = 0.5. Subsequently, UMAP dimensionality reduction was conducted based on the first 30 principal components, followed by cell clustering using the FindNeighbors and FindClusters functions with a resolution of 0.1. Expression of known marker genes based on the CellMarker 2.0 (http://bio-bigdata.hrbmu.edu.cn/CellMarker) database to determine cell type.

### Differential analysis and functional enrichment analysis

To investigate the heterogeneity in gene expression patterns between cell subpopulations, the FindAllMarkers function in Seurat was utilized to identify highly expressed genes for each cell subgroup, with parameters set as only.pos = T, min.pct = 0.25, and logfc.threshold = 0.25. The clusterProfiler package (version 4.8.2) (*Yu et al., 2012*) was used to examine the biological functions associated with the differentially expressed genes (DEGs) set, with parameters set to keyType = "SYMBOL", $p$valueCutoff = 0.05, and qvalueCutoff = 0.1.

## Pathway activity analysis and copy number variation analysis

TCGA-LIHC RNA-seq data were downloaded from The Cancer Genome Atlas (TCGA, https://portal.gdc.cancer.gov/) database, with expression values transformed to log2(fpkm +1). Survival time and status data were also retrieved. The enrichment scores for the MAPK signaling pathway gene set in TECs were computed using the AUCell package (*Aibar et al., 2017*). The inferCNV package's (*Li et al., 2022*) CreateInfercnvObject function was employed to create an inferCNV object for TECs, with B cells as the reference, and infercnv::run function was used for CNV analysis.

## Pseudotime analysis

Pseudotime trajectory analysis was conducted using Monocle (*Trapnell et al., 2014*). The FindMarkers function was applied to construct differentially expressed gene sets between groups, with parameters |log2FC| > 0.25 and min.pct = 0.25, aiding in the construction of differentiation trajectories. The newCellDataSet was used to create a cds object, low-quality cells were filtered out, and the DDRTree algorithm was implemented for dimensionality reduction using the reduceDimension function. The orderCells function sorted the cells, plot_cell_trajectory function visualized the trajectory, and plot_genes_in_pseudotime function displayed scatter plots showing the expression levels of CNV-affected genes in the MAPK pathway across pseudotime. The starting point of the differentiation trajectory was determined based on biological significance.

## Transcription factor target prediction

Transcription factors in HCC were predicted using JASPAR, an open-source database of transcription factor binding site information (*Castro-Mondragon et al., 2022*; *Fornes et al., 2020*). The JUN gene was searched for *Homo sapiens*, with default settings leading to the selection of JUN gene ID MA0489.1. The DNA sequence for the PGF gene (NG_029168.1) was retrieved from NCBI's Nucleotide database, selecting the sequence between 1 bp and 3,000 bp, with the Relative profile score threshold set to 85.

## Cell cultures and quantitative real-time PCR

The normal adult liver epithelial THLE-2 cells and human HCC-huh7 cells (composing of epithelial and tumorigenic cells with highly heterogeneity) were purchased from the COBIOER corp. (Nanjing, China). These cells were cultured in DMEM medium (Invitrogen, Waltham, MA,USA) that supplemented with 1% penicillin-streptomycin and 10% fetal bovine serum (FBS) in an incubator containing 5% $CO_2$ at 37 °C. Cells were sub-cultured 1–3 times at full confluence before experiments, the cell lines were regularly checked free from any mycoplasma contamination (*Bamodu et al., 2020*). The TRizol reagent (Invitrogen) was used for the total RNA extraction, then the PrimeScript Reverse Transcriptase (TaKaRa) was used for the cDNA synthesis, the LightCycler 480 (Roche) was used for the qPCR detection according to the manufacturer's specification. The reference genes GAPDH and the $2^{-\Delta\Delta Ct}$ method was used for the expression level analysis of interest genes, each sample performed three times biological and technique replicates. The specific primers are showed in Table S1.

## Trans-well and wound healing assay

For transient silencing of PGF, the specific si-PGF reagent was purchased from the Sigon corp. (Suzhou, China), the work concentration was configured for silenced cells following the manufacturer's protocol, the cells were used for the experiment after 48 h transfection. Cell invasion was analyzed by the trans-well assay, a total of $5 \times 10^4$ cells was seeded into the chamber (consisting of 8-μm pore membrane filter inserts) with 200 μL serum-free DMEM medium, the lower chamber includes 600 μL DMEM medium for 24 h, then the 4% paraformaldehyde for 15 min cell fixation and 0.1% crystal violet for staining were performed at room temperature, finally the inverted microscope was used for cell imaging (*Bamodu et al., 2020*). For the cell migration assessment, the wound healing assay were performed. First, HCC-huh7 cells were seeded into six-well plates with DMEM medium and cultured until confluency, then a rectilinear scratch was conducted by utilizing a 100 μL pipette tip. After 24 h incubation, the 4% paraformaldehyde and 0.1% crystal violet were used for 15 min cell processing orderly, an inverted microscope used for cell imaging (*Bamodu et al., 2020*).

## Statistical analysis

Continuous variables between two groups were compared using Student's t-test. All computational processes were performed using R language (version 4.3.1). A *p*-value of $< 0.05$ was considered statistically significant.

# RESULTS

## Single-cell mapping in HCC tissues

In the single-cell atlas of HCC tissue, data from 3,627 cells were retained for subsequent annotation analysis following preprocessing steps such as normalization, dimensionality reduction, and clustering. Six cell populations (including TECs, myofibroblast, B cells, TAMs, T cells, and liver bud hepatic cells) within the HCC tissue were identified based on the expression levels of marker genes. In TECs, significant upregulation of VEF, ENG, and CDH5 was observed. Liver bud hepatic cells exhibited pronounced expression of CPB2, MASP2, CFHR1, and UGT2B4. High expression levels of CD3E, CD3G, CD2, and CD3D were noted in T cells. Myofibroblasts showed significant upregulation of COL1A1 and COL3A1. Elevated expression of CD79A, SLAMF7, FCRL5, and MZB1 was detected in B cells. Tumor-associated macrophages (TAMs) demonstrated significant upregulation of CD163, CD68, and CSF1R (Figs. 1A–1C). A discrepancy in the numbers of the six cell populations was discovered, with a higher count of T cells, TECs, and liver bud hepatic cells, while the TAMs were the least prevalent, including 741 T cells, 726 TECs, 704 liver bud hepatic cells, 620 B cells, 523 Myofibroblasts, and 313 TAMs (Fig. 1D). Furthermore, the activity of the MAPK signaling pathway in these six cell populations was examined, revealing activation in TECs according to AUCells analysis results (Fig. 1E). These results reveal the complexity and heterogeneity of the HCC tumor microenvironment and that TECs may play a critical role in HCC growth and progression through the MAPK signaling pathway.

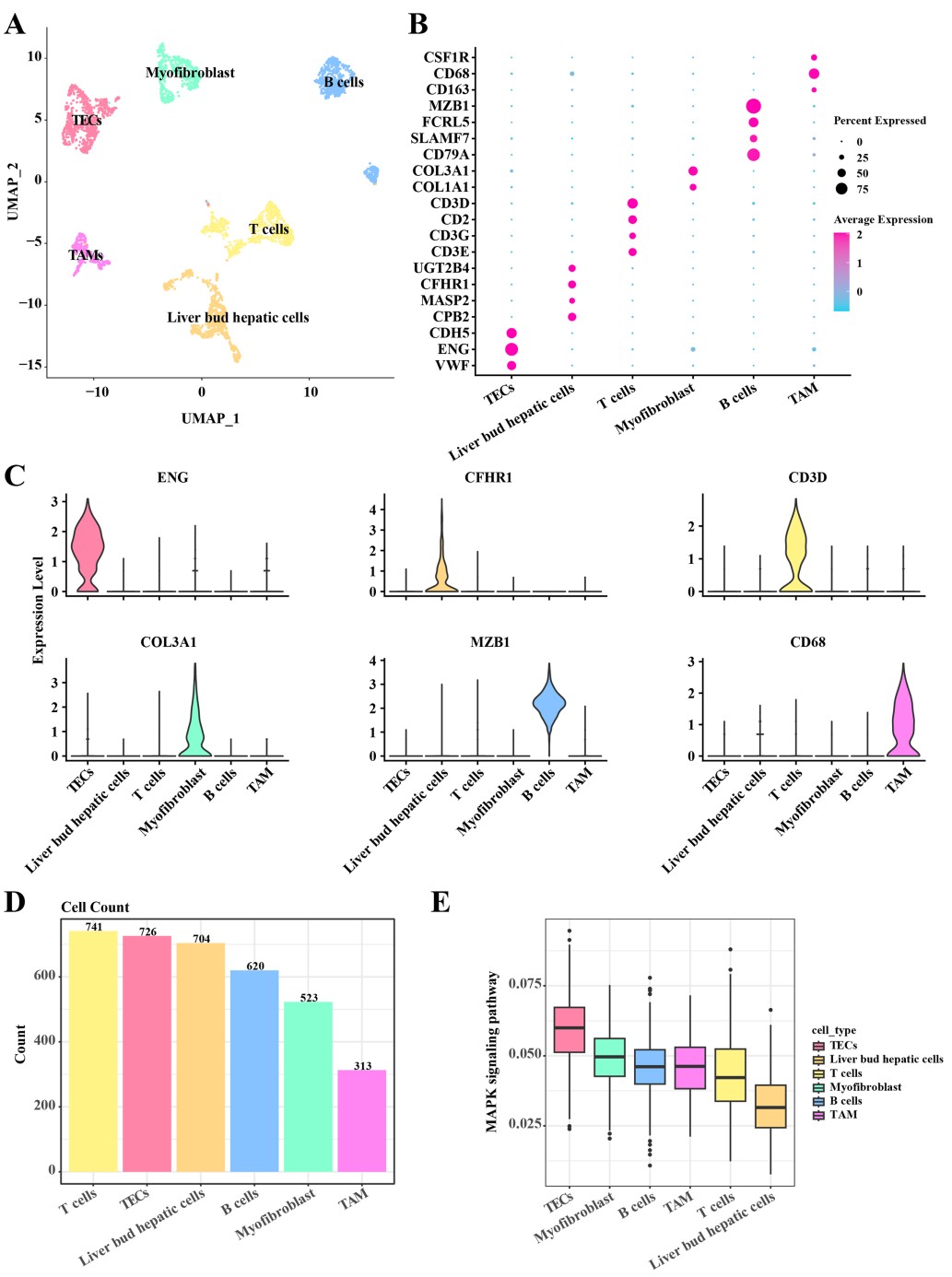

**Figure 1 Single cell atlas in HCC tissue.** (A) Distribution of the six cell populations after UMAP clustering. (B) Bubble plots of marker genes expressed in the six cell populations. (C) Violin plot of marker genes expressed in the six cell populations. (D) Histogram of cell counts. (E) Box plots of MAPK signaling pathway activity in the six cell populations.

## CNV analysis of TECs

Based on hidden markov model (HMM), predictions were conducted for 19 genomic regions in B cells and TECs. Significant variations were observed in TECs, whereas B cells

exhibited minimal genomic alterations. As shown in Fig. 2A, we observed that significant genomic variation was displayed in TECs, with many chromosomal regions exhibiting significant gene amplifications or deletions. For example, several regions on chromosomes 2, 7, 8, 12, 17, 20, and 22 exhibited significant amplifications and deletions. Subsequent analysis focused on the variations within the MAPK signaling pathway genes in TECs. Amplifications were identified in genes such as CDC42, ECS1T, HSPB1, MAP3K11, MAPK3K6, NRAS, and RCA1 (Fig. 2B), while deletions were noted in AKT1, MAP4K4, MAPK1, MAPK3, PDGFD, and PTPN7 (Fig. 2C). These findings suggested that amplifications and deletions of specific genes in HCC patients might be associated with cancer development.

## Single-cell landscape of TECs

Further normalization, dimension reduction, and clustering were applied to TECs to elucidate their heterogeneity, resulting in the identification of two cell subgroups: TECs 1 and TECs 2 (Fig. 3A). TECs 1, the predominant subgroup, accounted for approximately 83.61%, while TECs 2 constituted about 16.39% (Fig. 3B). Functional enrichment analysis revealed that TECs 1 were primarily enriched in biological processes like cytoplasmic translation, regulation of vasculature development, and angiogenesis. In contrast, TECs 2 were mainly associated with ameboidal-type cell migration, tissue migration, epithelial cell proliferation, and migration (Fig. 3C). Notably, TECs 1 were active in pathways such as fatty acid metabolism and cholesterol homeostasis, whereas TECs 2 were more engaged in the Wnt signaling pathway and interferon-alpha response (Fig. 3D). These results indicated that TECs 1 and TECs 2 might play distinct biological roles in the liver, with TECs 1 potentially more involved in metabolic processes and vascular development, while TECs 2 might be more closely related to cell migration and immune responses.

## CNV variants in MAPK pathway genes in TECs

To further understand the mechanisms of HCC progression, DEGs were identified in TECs 1 and TECs 2, intersecting with CNV variants in the MAPK signaling pathway. Fifteen DEGs were determined: DUSP6, FOS, GADD45B, HSPA1A, HSPA1B, HSPB1, JUN, JUND, NR4A1, PDGFRB, PGF, RASGRP3, RRAS, TGFBR2, and TNFRSF1A (Fig. 4A). Elevated expression levels were noted for DUSP6, HSPA1A, HSPA1B, HSPB1, RASGRP3, TGFBR2, and TNFRSF1A in TECs 1, while FOS, GADD45B, JUN, JUND, NR4A1, PDGFRB, PGF, and RRAS were predominantly expressed in TECs 2 (Figs. 4A, 4B). The CNV status of these 15 DEGs was showcased in TECs, with TNFRSF1A and HSPB1 exhibiting high amplification frequencies, approaching or reaching 100%. This indicated that the copy numbers of these genes were significantly elevated in TECs. Conversely, deletions such as those in GADD45B and NR4A1 were less common (Fig. 4C).

## Differentiation trajectory of TECs

Pseudotime analysis was conducted to explore the role of TECs in HCC progression, showing that TECs 1 served as the developmental starting point, transitioning towards TECs 2 as the endpoint (Fig. 5A). The expression of CNV-altered DEGs within the MAPK

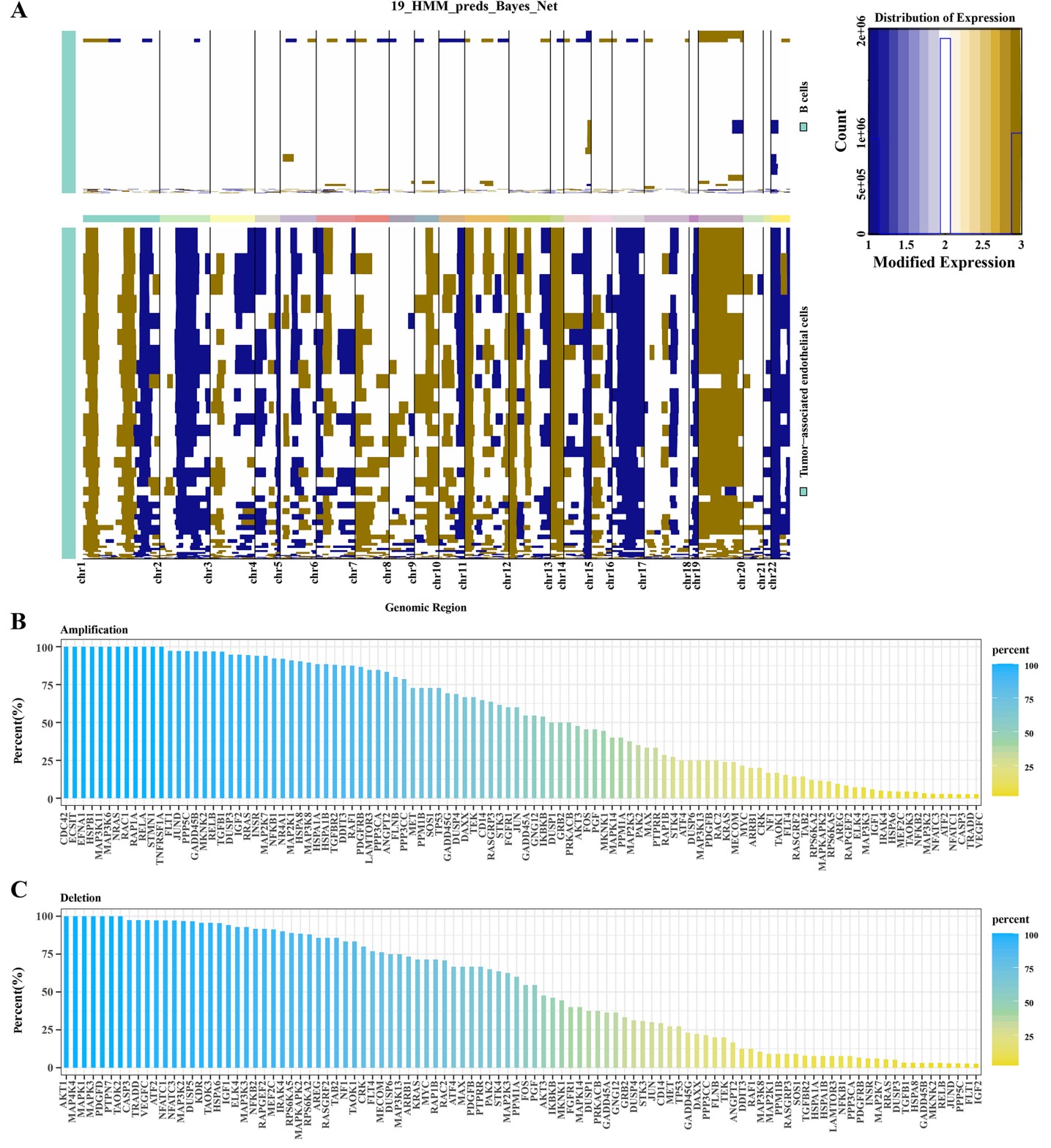

**Figure 2 CNV variation analysis in TECs.** (A) CNV variation spectra in B cells and TECs. (B) Variation rates of amplified genes in the MAPK signaling pathway in TECs. (C) Variation rates of deleted genes in the MAPK signaling pathway in TECs.

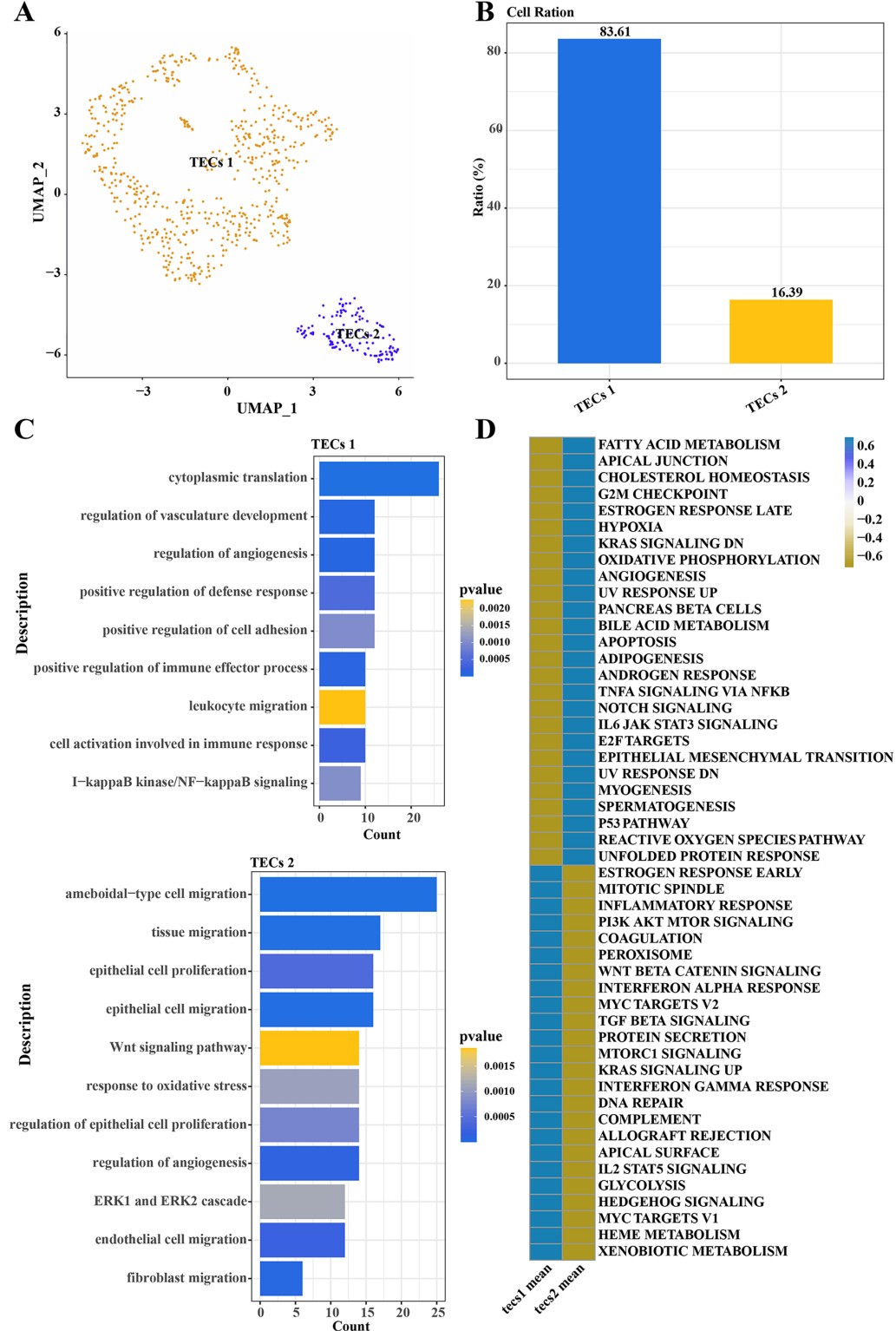

**Figure 3 Single-cell atlas of TECs.** (A) UMAP scatter plot showing the single-cell atlas of TECs, identifying two cell populations, TECs 1 and TECs 2. (B) Bar graph of the cell proportion of TECs 1 and TECs 2. (C) Biological processes involving TECs 1 and TECs 2. (D) AUC scores of Hallmark pathway activities in TECs 1 and TECs 2.

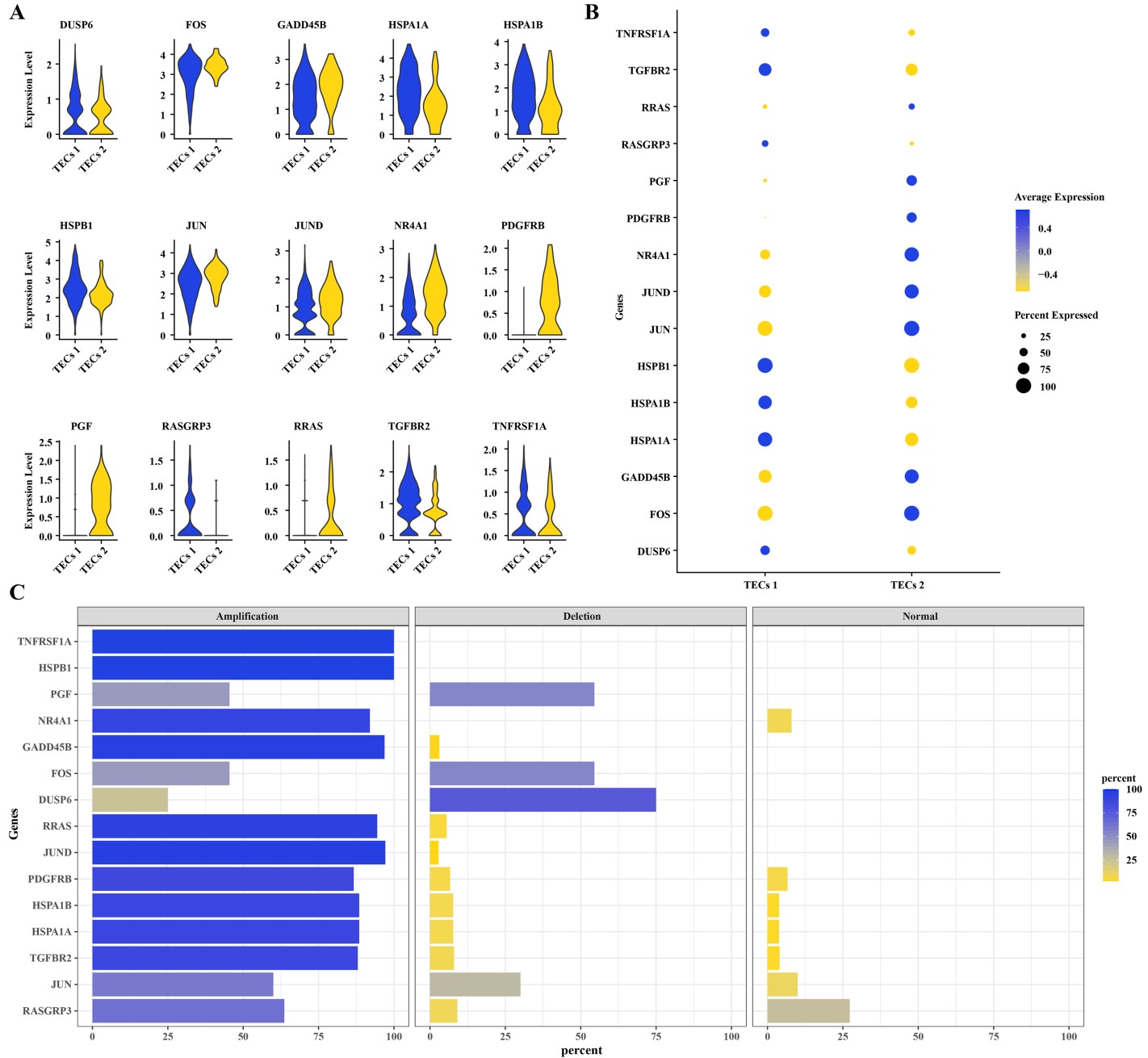

**Figure 4 CNV variants in MAPK pathway genes in TECs.** (A) Expression levels of 15 DEGs with CNV variations in the MAPK signaling pathway in TECs 1 and TECs 2. (B) Average expression levels of 15 DEGs with CNV variations in the MAPK signaling pathway. (C) Variation rates of 15 DEGs with CNV variations in the MAPK signaling pathway.

pathway, specifically PDGFRB, PGF, JUN, and NR4A1, was observed to consistently increase from the TECs 1 subgroup to TECs 2 (Fig. 5B). The prognostic value of these genes was assessed, revealing that high expression of JUN and PGF was associated with poor HCC prognosis (Figs. 5C, 5D). The JUN gene, part of the AP-1 transcription factor complex, forms heterodimers with FOS family proteins, enhancing DNA binding and transcriptional activity across various cell types (*Ji et al., 2012*). The PGF gene encodes a

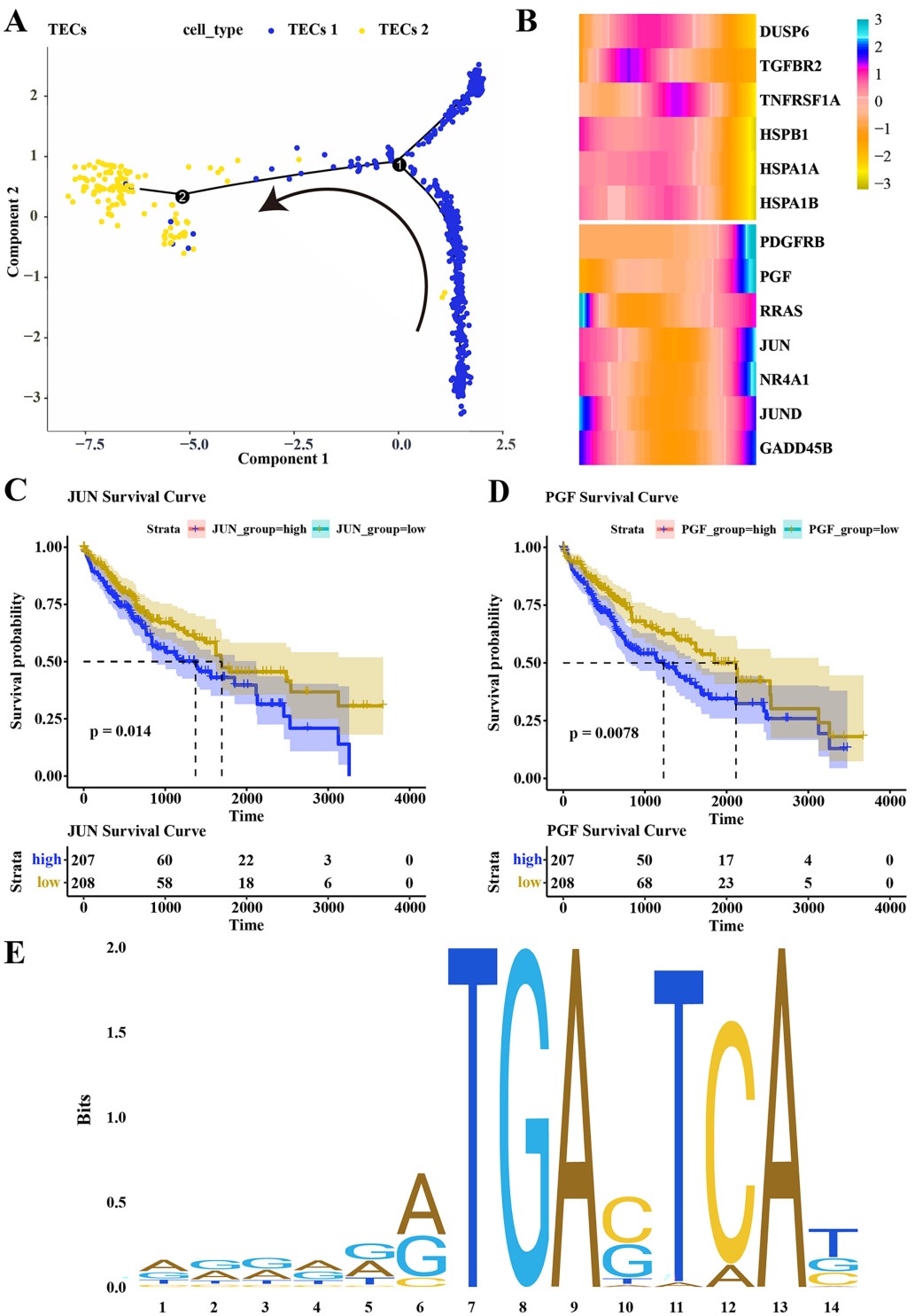

**Figure 5 Differentiation trajectory of TECs.** (A) Developmental trajectory of TECs, differentiating from TECs 1 to TECs 2. (B) Expression trends of DEGs with CNV variations in the MAPK signaling pathway during the development from TECs 1 subgroup to TECs 2 subgroup. (C) Survival analysis of patients grouped by JUN expression. (D) Survival analysis of patients grouped by PGF expression. (E) Binding sites of JUN transcription factor.           

**Table 1 JUN target PGF relative score.**

| Matrix ID | Name | Score | Relative score | Sequence ID | Start | End | Strand | Predicted sequence |
|---|---|---|---|---|---|---|---|---|
| MA0489.1 | MA0489.1.JUN | 15.406107 | 0.963182368 | NG_029168.1 | 936 | 949 | + | GGGAAGTGAGTCAG |
| MA0489.1 | MA0489.1.JUN | 11.13969 | 0.908281638 | NG_029168.1 | 1,001 | 1,014 | + | ACACGATGACTAAT |
| MA0489.1 | MA0489.1.JUN | 9.433627 | 0.886327823 | NG_029168.1 | 1,496 | 1,509 | + | ATTCAATGAGTCAA |
| MA0489.1 | MA0489.1.JUN | 7.3845983 | 0.859960688 | NG_029168.1 | 169 | 182 | + | GGCACGTGAGTAAG |
| MA0489.1 | MA0489.1.JUN | 7.0284495 | 0.855377726 | NG_029168.1 | 941 | 954 | − | GCTTCCTGACTCAC |
| MA0489.1 | MA0489.1.JUN | 7.0071077 | 0.855103098 | NG_029168.1 | 1,169 | 1,182 | − | AGGAGCTGACTCTT |

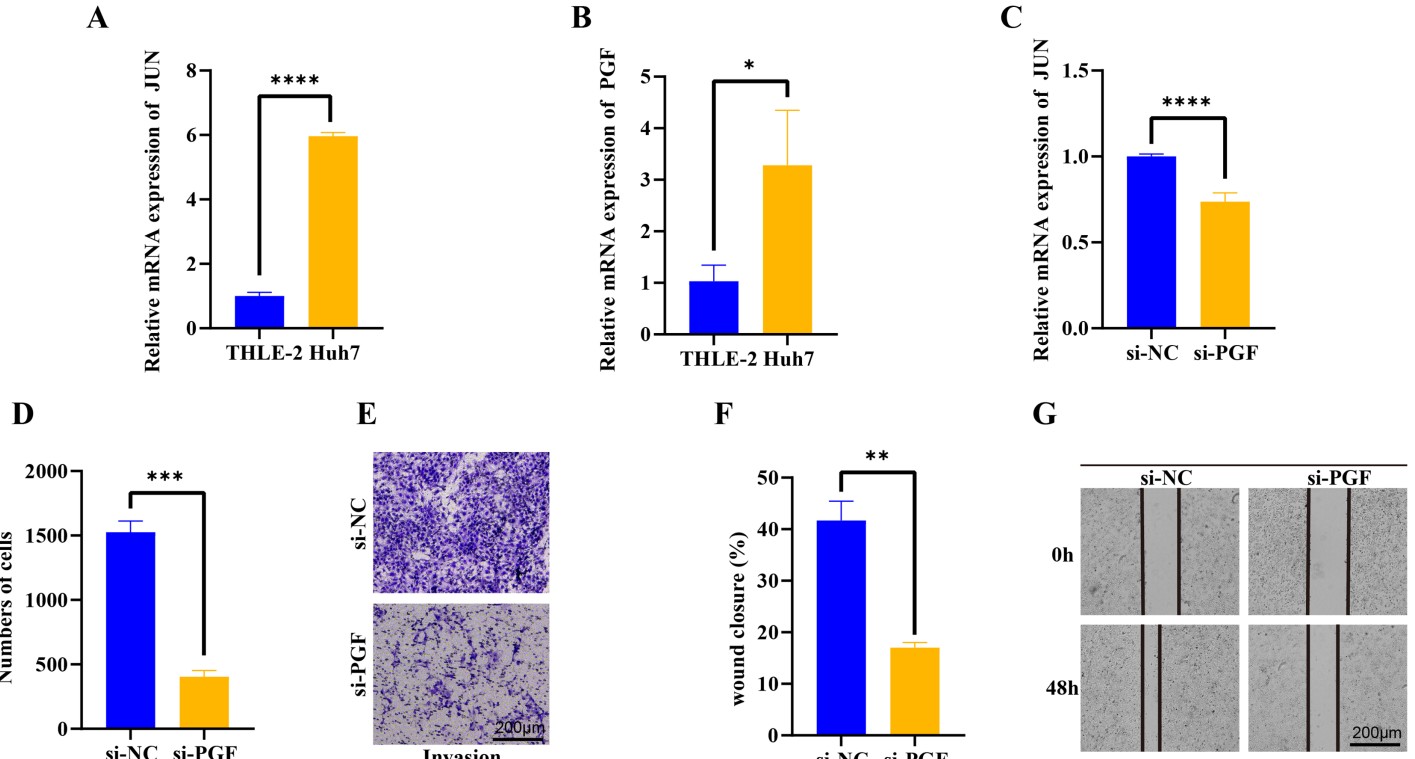

**Figure 6 qPCR, trans-well and wound healing assay analysis of interet genes.** (A) The basic expression of JUN gene in THLE-2 and Huh7 cells. (B) The basic expression of PGF gene in THLE-2 and Huh7 cells. (C) The expression of JUN gene in Huh7 cells with si-PGF silencing. (D) The number of Huh7 cells with si-PGF silencing. (E) The trans-well analysis of Huh7 cells with si-PGF silencing. (F) Analysis of wound closure percent of Huh7 cells with si-PGF silencing. (G) The wound healing analysis of Huh7 cells with si-PGF silencing. *$p < 0.05$, **$p < 0.001$, ***$p < 0.001$, ****$p < 0.0001$.

member of the placental growth factor (PlGF) protein family, playing roles in angiogenesis and endothelial cell growth by stimulating their proliferation and migration, and interacting with receptors FLT1 and VEGFR-1. PlGF-2 binds to neuropilins NRP1 and NRP2 in a heparin-dependentmanner (*Van Bergen et al., 2019*; *Vrachnis et al., 2013*; *Nejabati et al., 2017*). Consequently, JUN is predicted to regulate PGF expression, potentially promoting the migration of liver cancer cells. The JAPAR online tool was employed to predict the binding sites of the JUN gene on the PGF gene (NG_029168.1), focusing on the 1–3,000 bp region of the DNA sequence, identifying binding sites with

high similarity scores at positions 936–949 and 1,001–1,014, suggesting JUN's regulatory role over PGF expression (Fig. 5E; Table 1). These findings contribute to a better understanding of the complex biology of HCC and may offer new targets for therapeutic intervention, highlighting the distinct roles that TECs subgroups may play in liver function and disease progression.

## PGF as key regulatory factor mediated cell proliferation and migration

We further evaluated the JUN and PGF function in HCC progression. In the Huh7 tumor cells, the JUN and PGF expression were significantly up-regulated and the JUN exhibited a more significant increase ($p < 0.05$, Figs. 6A, 6B), indicating that these genes played an important role in promoting cancer progression. After the PGF silencing, the proto-oncogene JUN expression level was significantly inhibited ($p < 0.05$, Figs. 6C). Further invasion and migration assay showed that the invaded cells in si-PGF group were obviously reduced compared with the si-NC control ($p < 0.05$, Figs. 6D, 6E), the wound closure percent was significantly decreased in si-PGF group more than 1-fold ($p < 0.05$, Figs. 6F, 6G), these results suggested that the PGF may be characterize the differentiated aggressive HCC cells and mediated cell proliferation and migration in cancer progression.

## DISCUSSION

In the study of HCC, single-cell RNA sequencing technology has provided in-depth insights into the complexity of the tumor microenvironment. This approach enabled researchers to unveil cellular heterogeneity within liver cancer, identifying distinct cellular subpopulations and examining their roles in tumor progression. The research in single-cell RNA sequencing for HCC highlighted the heterogeneity and immunosuppressive landscape within the tumor microenvironment. Analysis of single-cell samples from HCC tissues revealed various cell populations, including endothelial cells, hepatic progenitor cells, T cells, myofibroblasts, B cells, and tumor-associated macrophages (TAMs), annotated with specific marker genes (Xie et al., 2023; Ho et al., 2021). A study analyzing HBV-related HCC uncovered a potential inverse correlation between tumor-infiltrating T cells and TAMs. Furthermore, an enrichment of immune-suppressive molecules was observed in TAMs, significantly correlating with poorer survival rates in HCC patients (Ho et al., 2021; Dou et al., 2022). Additionally, single-cell RNA sequencing exposed potential immunotherapeutic targets within liver cancer, essential for understanding the tumor microenvironment's complexity and heterogeneity. Precise cell type annotation revealed the specific roles cells play in tumor progression, potentially guiding the development of future therapeutic strategies (Xie et al., 2023). Moreover, the integration of single-cell data with TCGA data enabled researchers to assess the correlation between specific cell types' proportions and HCC and its survival outcomes, aiding in identifying cell types associated with HCC and understanding their role in disease progression and patient prognosis (Alvarez et al., 2022). The study also identified six cellular populations within HCC tissues, annotated and recognized through specific marker genes. These data not only unveiled the distribution of cell types but also provided clues to understanding the specific functions of cells within the tumor.

Particularly in TECs, CNVs were found that might be associated with cancer progression. Research indicated that specific CNV-driven genes could be utilized to construct prognostic models for HCC, associated with the infiltration of immune cells in the tumor microenvironment. Specifically, we found significant amplifications or deletions in multiple genomic regions of total TECs and copy number changes in genes in important MAPK signaling pathways associated with cancer progression. For instance, gains and losses in copy number primarily affected genes in chromosome regions 1 q and 8 p, related to several pathways associated with HCC progression. These regions might play a crucial role in the progression of HCC, particularly focusing on genes such as YY1AP1 and CHMP7 might be particularly crucial (*Zhou et al., 2017*; *Shahrisa et al., 2021*). Moreover, a series of CNV-driven genes were discovered, significantly correlated with the overall survival of HCC patients and related to the characteristics of immune cells infiltrating the tumor microenvironment, potentially aiding in improving the understanding of HCC's driving mechanisms and offering an immunological perspective for personalized treatment (*Bian et al., 2021*). Variations within genes of the MAPK signaling pathway in TECs, such as amplifications in CDC42, HSPB1, NRAS, and deletions in AKT1, MAPK1, might reveal their significance in HCC, possibly related to their roles in tumor growth and progression, particularly in signaling pathways. In summary, CNV studies in HCC suggest that specific CNV variations are closely associated with liver cancer development, potentially offering new targets and biomarkers for the treatment and prognosis of HCC.

Our findings demonstrated that tumor endothelial cells (TECs) could be divided into two subgroups, TECs 1 and TECs 2, based on the expressed marker genes, exhibiting distinct biological processes and signaling pathway activities in functional enrichment analyses. The differentiation of TECs may indicate that different subgroups of endothelial cells play varying roles in different stages or subtypes of HCC. The role of cytoplasmic translation has been extensively studied in cancer progression and treatment resistance. Alterations in cytoplasmic translation are closely related to the phenotypic plasticity, survival capabilities, and treatment responses of cancer cells. Studies have shown that cancer cells can adapt to adverse growth conditions, such as nutrient deprivation or therapeutic stress, by modulating the translation process (*Fabbri et al., 2021*; *Robichaud et al., 2019*). The regulation of vasculature development and angiogenesis is crucial for tumor growth and metastasis. The formation of tumor vasculature not only supplies the tumor with essential nutrients and oxygen but also facilitates the invasion and distant metastasis of tumor cells (*Li et al., 2019*; *Lugano, Ramachandran & Dimberg, 2020*). Therefore, the TECs 1 subgroup may play a significant role in the angiogenesis of HCC, potentially serving as a therapeutic target, particularly in inhibiting tumor angiogenesis. The biological processes enriched in the TECs 2 subgroup, such as cell migration and epithelial cell proliferation, are essential for the invasiveness and metastatic potential of tumors. These processes enable the tumor to breach the basement membrane, enter the bloodstream, and form metastatic foci at distant body sites. Hence, the functions of TECs 2 may play a crucial role in the invasiveness and metastasis of HCC. In summary, the functional differences between the TECs 1 and TECs 2 subgroups in HCC reveal their potential distinct roles in tumor biology, providing potential targets and biomarkers for

future HCC treatment strategies. These findings underscore the importance of further investigating the heterogeneity of TECs in HCC to better understand their role in tumor progression and to develop treatments targeting these processes.

Pseudotime analysis revealed the dynamic changes in TECs subgroups during tumor progression, with TECs 1 serving as the starting point and TECs 2 as the endpoint, suggesting a possible sequential transition in disease development. The transition from TECs 1 to TECs 2 may reflect the dynamic changes of cells within tumor progression. This transformation could be associated with the expression levels of genes such as PDGFRB, PGF, JUN, and NR4A1, which play crucial roles in tumor development, angiogenesis, and cell proliferation (*Gao et al., 2023*). Specifically, the expression of JUN and PGF genes increased during the transition from TECs 1 to TECs 2 and was associated with poor prognosis in HCC. JUN, part of the AP-1 transcription factor, is closely related to cell growth and differentiation, while PGF, a placental growth factor, plays roles in angiogenesis and cell proliferation in several cancers, including HCC (*Ji et al., 2012*; *Kim et al., 2023*). High expression of PDGFRB has also been shown to negatively affect the prognosis of gastric cancer patients by promoting angiogenesis and modulating the tumor immune microenvironment (*Liu et al., 2022*). In addition, *Wu et al. (2017)* found that NR4A1 was downregulated in TNBC and that restoration of NR4A1 expression inhibited the growth and metastasis of triple-negative breast cancer cells, suggesting that NR4A1 is a tumor suppressor in this disease. The changes in the expression of these genes may reflect the role of TECs in the progression of HCC, especially in modulating the tumor microenvironment and promoting tumor cell behaviors. Although direct literature on pseudotime analysis of TECs in HCC is limited, these analytical results align with extensive HCC studies exploring cellular heterogeneity in the tumor microenvironment, molecular mechanisms of tumor progression, and potential biomarkers and therapeutic targets (*Balogh et al., 2016*). In conclusion, by integrating pseudotime analysis with existing literature, we can gain deeper insights into the role of TECs in HCC, particularly how they participate in tumor progression through dynamic changes in gene expression. These insights provide valuable information for future research directions and potential therapeutic strategies. Although we demonstrated that the PGF as key regulatory factor mediated cell proliferation and migration, whether the specific molecular events (such as the apoptosis or cell cycle control) that are PGF involved, is need to be further elucidated.

In summary, this study intricately reveals the heterogeneity of TECs in HCC and their critical roles in tumor progression, particularly through elucidating the functions of differentially expressed genes associated with the MAPK signaling pathway. These findings not only enhance our understanding of cellular heterogeneity within the HCC microenvironment but also provide potential molecular targets for future therapeutic strategies.

## ABBREVIATIONS

**TECs**    tumor endothelial cells
**HCC**    hepatocellular carcinoma
**GEO**    Gene Expression Omnibus

| | |
|---|---|
| **UMAP** | Uniform Manifold Approximation and Projection |
| **CNV** | copy number variations |
| **TCGA** | The Cancer Genome Atlas |
| **DEGs** | differentially expressed genes |
| **PlGF** | placental growth factor |

### Funding

This study was supported by the Science and technology Research program of Henan Province (No. 242102311156). The funders had no role in study design, data collection and analysis, decision to publish, or preparation of the manuscript.

### Grant Disclosures

The following grant information was disclosed by the authors:
Science and Technology Research Program of Henan: 242102311156.

### Competing Interests

The authors declare that they have no competing interests.

### Author Contributions

- Jiachun Sun conceived and designed the experiments, analyzed the data, authored or reviewed drafts of the article, and approved the final draft.
- Shujun Zhang conceived and designed the experiments, prepared figures and/or tables, and approved the final draft.
- Yafeng Liu conceived and designed the experiments, analyzed the data, prepared figures and/or tables, and approved the final draft.
- Kaijie Liu performed the experiments, prepared figures and/or tables, authored or reviewed drafts of the article, and approved the final draft.
- Xinyu Gu performed the experiments, analyzed the data, authored or reviewed drafts of the article, and approved the final draft.

### Data Availability

  The public dataset used in this study is available at Genbank: GSE125449.
  The raw data is available in GitHub and Zenodo:
  - https://github.com/123xinyugu/Raw-data.git
  - 123xinyugu. (2024). 123xinyugu/Raw-data: Raw data (v.1.1.0). Zenodo. https://doi.org/10.5281/zenodo.12716703.
  https://www.ncbi.nlm.nih.gov/geo/query/acc.cgi?acc=GSE125449

### Supplemental Information

Supplemental information for this article can be found online at http://dx.doi.org/10.7717/peerj.18362#supplemental-information.

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
