# Peer review of "Exploring tumor endothelial cells heterogeneity in hepatocellular carcinoma: insights from single-cell sequencing and pseudotime analysis"

_PeerJ, doi:10.7717/peerj.18362_

## Round 0.1 · original submission · Major Revisions

We have now received comments from two expert reviewers. Both reviewers have recommended major revisions to your manuscript. After thorough consideration, I concur with their assessment and have decided that your manuscript requires major revisions before it can be considered further for publication. Please carefully address all the concerns and suggestions raised by the reviewers. In particular, pay close attention to:

1. Strengthening the methodology section and providing more detailed explanations of your experimental procedures.
2. Expanding the discussion to more thoroughly interpret your results in the context of existing literature.
3. Improving the clarity and quality of your figures and tables.
4. Addressing any inconsistencies or errors pointed out by the reviewers.

We believe that your work has potential, but significant improvements are necessary to meet the standards of our journal. Please provide a point-by-point response to all reviewers' comments along with your revised manuscript.

Reviewer 1 ·

Basic reporting

In this study, the author explores the potential link between the tumor endothelial cells heterogeneity and the development and prognosis of hepatocellular carcinoma (HCC). Prognostic factors in hepatocellular carcinoma were identified through bioinformatics methods. The experimental design is rigorous.
1. In the results of abstract section, what is the functional activities and signaling pathways difference among two subgroups.
2. In abstract, the description of the method is too detailed and the description of the result is too brief.
3. Line 100-102, What method is used for cell type annotation, and what is the reference gene set.
4. Line 113, What is the purpose of getting an MAPK signaling score. What kind of indicator does it exist as.
5. Line 118-119, the DEGs of this section whether is consistent with the Line 104-107.
6. Line 129, What is the JUN gene, why search for it as an important target gene.
7. Line 170-171, the six cell populations included what. Is this was showed in Figure 1A. Please complete it.
8. Line 185, What does HMM refer to. It appears only once in the full text, and there is no acronym explanation.

Experimental design

no comment

Validity of the findings

9. Line 186-187, the description of Figure 2A results is vague, what is the significant variations in TECs. The chromosomal variation characteristics should be appropriately described.
10. Line 187-192, Is MAPK a key cancer pathway. Is its mutation critical for cancer development, and has it been reported, please add to the discussion as appropriate.
11. Line 197, the Functional enrichment analysis of two subgroups were performed, meanwhile, the DEGs of two groups were performed in Line 209, Why is the CNV state of DEGs explored here instead of the functional enrichment analysis of these DEGs. CNV can generally analyze the mutation frequency of genes. But there is no frequency mutation shown in this result.

Reviewer 2 ·

Basic reporting

This study aimed to explore the heterogeneity of cellular subpopulations in hepatocellular carcinoma (HCC) by analyzing public databases and to reveal the regulatory role of the interactions of these cellular subpopulations for HCC progression. This study firstly distinguished different cell subpopulations by single-cell clustering of HCC data samples from public databases. Next, the differentiation trajectories, copy number variability of these cells were analyzed and revealed that the downstream pathways of endothelial cells could be potential factors affecting HCC progression. This study also incorporates cellular experiments to investigate the regulatory role of marker genes of key cell subpopulations in cancer cell proliferation and migration. In conclusion, this study integrates bioinformatics analyses and cellular experiments and is a comprehensive study, but the following issues still need to be addressed before publication:
1. The abstract section of this study does not fully elaborate the bioinformatics analysis methodology of this paper, especially by what means the cell subpopulations are clustered, and what are the key marker genes for these cell subpopulations are not clearly elaborated in the abstract. Please complete the abstract.
2. Since this study analyzed the potential role of downstream regulatory pathways of cellular subpopulation marker genes in the progression of HCC, why did it not perform further gene expression analyses to confirm the tangible regulatory role of these pathways? Please explain the description.
3. Heterogeneity of HCC is a systematic topic that used to be a research hotspot in the field of tumor therapy, but relevant studies have not been common in recent years, and thus what question is the starting point of this paper to conduct this research? Please add the starting point of the study in this paper.
4. What is the inter-regulatory relationship between PDGFRB, PGF, JUN and NR4A1 genes and the dynamics of HCC? Please add a description in the abstract section and elaborate clearly in the results section.
5. The results section of this paper is not described clearly enough, for example, it is only stated that CNV analysis can reveal that TECs can influence HCC progression, but it is not stated what aspects of characterization were used to reach this conclusion, and it is recommended that this be supplemented and improved.
6. The introduction part does not state the intention of this paper clearly, and it is suggested to reorganize the language to make it clear, for example, what are the difficulties faced by the current therapies for HCC? How does the heterogeneity of the tumor microenvironment inspire the treatment of HCC? Are the findings of this paper expected to cope with the heterogeneity of HCC and alleviate the clinical symptoms of HCC patients?
7. Since this study points out that single-cell transcriptome analysis technology can help researchers to discover mechanisms of cancer progression, biomarkers, it is recommended to elaborate on this, especially to cite a few typical reports elucidating the role of single-cell transcriptome technology in molecular marker discovery in HCC.
8. The description in Figure 1 is not specific enough, please add a hypothetical description in the results section to illustrate how the differences in the number of the six cell populations reveal the characteristics of the HCC tumor microenvironment by proposing a plausible hypothesis based on the existing results.
9. The discussion section of this paper is mostly a pile-up of existing literature reports, which does not systematically elaborate the conclusions of this paper and does not highlight the innovativeness of this paper. Thus, it is recommended to deepen the discussion section in conjunction with the conclusions of this paper, for example, it is recommended to describe PDGFRB, PGF, JUN, NR4A1 and other genes in conjunction with the characterization of their molecular pathways.
10. lines 323-324 illustrate that PGF is a placental growth factor that plays a role in angiogenesis and cell proliferation, but in which cancer types does this gene mediate angiogenesis and cell proliferation? Do these cancer types include HCC?Please be specific about this literature to highlight the completeness of this article.

Experimental design

no comment

Validity of the findings

no comment

Additional comments

no comment

---

## Round 0.2 · accepted · Accept

Both reviewers have evaluated your revised manuscript and found that it adequately addresses their previous concerns. They have recommended acceptance of your paper in its current form. Thank you for your valuable contribution to our journal.

Reviewer 1 ·

Basic reporting

no comment

Experimental design

no comment

Validity of the findings

no comment

Additional comments

In this study, the authors explored the potential association between tumor endothelial cell heterogeneity and the development and prognosis of hepatocellular carcinoma (HCC). The prognostic factors of hepatocellular carcinoma were identified using bioinformatics methods and validated through in vitro experiments. The overall experimental design was rigorous, the methodology was well organized, the preface was rich, the discussion was in-depth, and the statistics were appropriate. The quality of the revised manuscript was significantly improved, seemingly meeting peerj's publishing standards.

Reviewer 2 ·

Basic reporting

The author first distinguished different cell subgroups by performing single-cell clustering on HCC data samples in public databases. Next, they analyzed the differentiation trajectory and copy number variability of these cells and found that downstream pathways of endothelial cells may be potential factors affecting HCC progression. In addition, they also combined cell experiments to study the regulatory role of key cell subpopulation marker genes in cancer cell proliferation and migration. In summary, the author attempts to explore the heterogeneity of hepatocellular carcinoma (HCC) cell subpopulations by analyzing public databases and reveal the regulatory role of these cell subpopulation interactions in HCC progression. As of the current version, it is impeccable and they have addressed all the reviewers' questions. This is a classic wet dry combination study, congratulations to the authors.

Experimental design

no comment

Validity of the findings

no comment